# A generalized framework for estimating snakebite underreporting using statistical models: A study in Colombia

**Carlos Bravo-Vega** [1] *, **Camila Renjifo-Ibañez**[2☯], **Mauricio Santos-Vega**[1,3☯], **Leonardo Jose León Nuñez**[4], **Teddy Angarita-Sierra**[5], **Juan Manuel Cordovez**[1]

**1** Grupo de investigación en Biología Matemática y Computacional (BIOMAC), Departamento de Ingeniería Biomédica, Universidad de los Andes, Bogotá, Colombia, **2** Corporación colombiana de investigación agropecuaria, AGROSAVIA, Bogotá, Colombia, **3** Facultad de Medicina, Universidad de los Andes, Bogotá, Colombia, **4** Observatorio de Salud Pública y epidemiología "José Felix Patiño", Universidad de los Andes, Bogotá, Colombia, **5** Grupo de investigación Biodiversidad para la sociedad, Universidad Nacional de Colombia sede de La Paz, Cesar, Colombia

☯ These authors contributed equally to this work.

* ca.bravo955@uniandes.edu.co

**Data Availability Statement:** All data and code used in this study is deposited in figshare repository (DOI: 10.6084/m9.figshare.21779342).

## Abstract

### Background

Snakebite envenoming is a neglected tropical disease affecting deprived populations, and its burden is underestimated in some regions where patients prefer using traditional medicine, case reporting systems are deficient, or health systems are inaccessible to at-risk populations. Thus, the development of strategies to optimize disease management is a major challenge. We propose a framework that can be used to estimate total snakebite incidence at a fine political scale.

### Methodology/Principal findings

First, we generated fine-scale snakebite risk maps based on the distribution of venomous snakes in Colombia. We then used a generalized mixed-effect model that estimates total snakebite incidence based on risk maps, poverty, and travel time to the nearest medical center. Finally, we calibrated our model with snakebite data in Colombia from 2010 to 2019 using the Markov-chain-Monte-Carlo algorithm. Our results suggest that 10.19% of total snakebite cases (532.26 yearly envenomings) are not reported and these snakebite victims do not seek medical attention, and that populations in the Orinoco and Amazonian regions are the most at-risk and show the highest percentage of underreporting. We also found that variables such as precipitation of the driest month and mean temperature of the warmest quarter influences the suitability of environments for venomous snakes rather than absolute temperature or rainfall.

### Conclusions/Significance

Our framework permits snakebite underreporting to be estimated using data on snakebite incidence and surveillance, presence locations for the most medically significant venomous

Data and code can be accessed here (https://figshare.com/articles/online_resource/Code/21779342).

**Funding:** This study was partially supported by: Minciencias, Colombia (https://www.minciencias.gov.co/), Application 727 for doctoral student to CBV, and Universidad de los Andes, Colombia (https://uniandes.edu.co/), Funding program for doctoral students, awarded to CBV. The funders had no role in study design, data collection and analysis, decision to publish, or preparation of the manuscript.

**Competing interests:** The authors have declared that no competing interests exist.

snake species, and openly available information on population size, poverty, climate, land cover, roads, and the locations of medical centers. Thus, our algorithm could be used in other countries to estimate total snakebite incidence and improve disease management strategies; however, this framework does not serve as a replacement for a surveillance system, which should be made a priority in countries facing similar public health challenges.

## Author summary

Snakebite envenoming is a neglected tropical disease that is a major challenge to manage because of incidence underreporting, which is caused by the low coverage of health centers in tropical countries and preferences for traditional medicine among snakebite victims. Several mathematical approaches have been used to explain variation in snakebite incidence via parameter adjustment, but these approximations have not been used to estimate envenomings that are not reported to medical centers. Our study proposes a statistical framework to estimate snakebite underreporting. To our knowledge, this is the first paper that provides computational estimates of the number of snakebite cases that are not receiving treatment by health care facilities; this approach is more time and labor-efficient compared with field epidemiological studies. Our modelling scheme can be used to rapidly estimate spatial heterogeneity in snakebite underreporting and identify areas where increased accessibility to health care facilities is urgently needed. Our framework can also be used to determine the expected number of snakebite cases, promote community empowerment and public health, and determine the real demand for antivenom. This tool could also be applied in various countries, but it is worth emphasizing that it should not serve as a replacement for medical surveillance system.

## Introduction

Snakebite envenoming is a neglected tropical disease (NTD) with high mortality and morbidity rates [1,2]. The most effective treatment is the administration of antivenom, which is composed of a mixture of antibodies [3]. Although antivenom is life-saving, most patients affected by a snakebite live in remote rural areas where it is not readily available [4]. In addition, poverty is known to affect envenoming risk and underreporting [5,6]. Envenomation episodes in rural areas often result in patients seeking traditional healers instead of antivenom therapy [2,7,8]. This poses a challenge for collecting data on snakebite incidence, which usually does not account for the total number of envenoming cases [9,10]. As a part of a global effort to reduce snakebite mortality by 50% in 2030, the World Health Organization (WHO) proposed a global strategy that involves strengthening surveillance systems, improving antivenom availability, and classifying this NTD as a major public health problem. The first step to achieving this objective is generating reliable estimates of snakebite burden, and disease eco-epidemiological approaches can help identifying overlooked vulnerable populations [11,12].

 Estimating disease burden by performing cross-sectional studies requires resources that are not readily available [13,14]. This necessitates studying the natural history of species associated with this NTD and mathematical modeling to estimate epidemiological parameters [15–17]. Such knowledge is critically important in megadiverse countries such as Colombia, which features approximately 319 snake species, but only 52 pose potential human health risks. Of these 52 species, 21 belong to the family *Viperidae* and 31 belong to the family *Elapidae* [18,19].

These venomous snakes are distributed throughout most of the country, and human–snake encounters occur frequently [20,21]. The most medically important venomous snake species in the country are *Bothrops asper* and *B. atrox*, as they are responsible for most of the envenomings (50–80%) [22,23]. Previous studies in Colombia have shown that *B. asper* occurs in the Caribbean, Pacific coast, and the inter-Andean valleys and that *B. atrox* occurs in the Orinoco and Amazonian regions [19]. However, few studies to date have examined the distributions of these two medically significant species and the effects of various environmental variables on their distributions [24,25].

Reliable and long-term data on snakebite envenoming that account for the number of cases among regions are usually lacking [8]. In Colombia, snakebite cases treated in medical centers are reported to the Sistema Nacional de Vigilancia en Salud Pública (SIVIGILA), which monitors and collects public health data [26]. In 2004, medical centers were required to report total snakebite cases, but in 2008, the reporting of each individual snakebite case became mandatory [22]. Snakebite incidence data report prior to 2004 were obtained from 1975 to 1999, with an average of 0.18 envenomings per 100.000 inhabitants nationwide (70.8 cases per year) [27]. In 2005, SIVIGILA reported 5.03 envenoming's per 100.000 inhabitants (2161 cases per year); in 2016, the reported incidence was 9.6 envenomings per 100.000 inhabitants (4704 cases per year). This sudden increase in cases appears to be associated with the implementation of the new reporting system rather than a change in the dynamics of snakebite [26].

Nevertheless, the total number of reported cases in Colombia is still expected to be less than the total number of cases due to complexities in data collection in the territory [22,28]. In this country, the costs of antivenom are paid by the government, but it often does not get distributed to areas most in need of this antivenom [22,29]. It is known that patients that do not have access to public health care are prone to use traditional medicine, so case-reporting do not capture these cases [2,8]. Thus, it is expected that snakebite underreporting exists in data published by SIVIGILA, even so the reporting system have improved significantly [22]. Therefore, estimating disease burden through mathematical approaches could help determine the total number of snakebite cases in the country and the real demand for antivenom, which would help achieve the goals of the WHO's global strategy [11,17,30].

Our article describes an algorithm to estimate underreporting using Colombia as a case study. For this, we developed a model that considers the risk of snakebites based on the distributions of the most important species from the epidemiological point of view. We tested its performance using the incidence registered in the country's public health system. Using a generalized model incorporating this risk estimator and other factors associated with snakebite underreporting, we estimated the spatial distribution of snakebite cases underreporting.

## Methodology

We followed three steps to generate and calibrate our model: First, we built an envenoming risk score map based on snake distributions (obtained through ecological niche modeling) and the law of mass action. Second, we calculated an accessibility score that reflects the cost of reaching a medical center. Finally, we developed and parametrized a mixed mathematical model to estimate underreporting.

### Snakebite's envenoming risk estimation

The envenoming snakebite risk is mainly associated with the presence of venomous snakes. Therefore, we developed an estimator of the snakebite envenoming risk score based on niche modelling for the two species. We referred to this risk score hereafter as *Snake niche modeling score* (*SNMS*, see below). To validate the risk score, we used the law of mass action, which

states that envenoming will be the result of encounters between humans and snakes, and it will be proportional to the multiplication of their abundances [17]:

$$I = \alpha \times S \times V \tag{1}$$

In Eq 1, $I$ is the snakebite incidence rate (Number of new cases over a period of time), $S$ is the rural population, $V$ is snake abundance, and $\alpha$ is the effective bite rate between both populations. This bite rate is a constant parameter, and it is obtained after merging parameters $\theta$ (probability that an encounter ends in a snakebite) and $\beta$ (contact rate between venomous snakes and humans) found in Eq 1 in [17]. After dividing Eq 1 by $P$, approximating $S$ to $SNMS$, and using a logarithmic transformation on the reported risk to linearize its relationship with $SNMS$, we can validate $SNMS$ by comparing it with the logarithm of reported person-time snakebite incidence rate using the Pearson's correlation coefficient (Eq 2):

$$Ln\left(\frac{I}{P}\right) \approx Ln(Person - time\ snakebite\ incidence\ rate) \approx a \times SNMS \tag{2}$$

## Snake presence data

We selected *B. asper* and *B. atrox* as study species because of their wide range, ability to adapt to human intervention, and the fact that they are the cause of most snakebites in Latin America and Colombia [19,25,31]. First, we built a database of presence records using georeferenced samples obtained from natural history museums in Colombia (see acknowledgments); duplicated records for each species were eliminated. To homogenize the spatial data, we used a 5 km x 5 km grid for both species so that there was only one presence point per square grid cell. The resolution of the grid was based on previous studies that have estimated the distributions of venomous snakes' in the Neotropics [15]. Finally, we removed records above 1900 m.a.s.l. for *B. asper* and 1500 m.a.s.l. for *B. atrox* because these are the maximum recorded altitudes known for each of these species in South America [19,32]. Our initial snake presence database had a total of 636 records for *B. asper* and 374 records for *B. atrox*. After data depuration, our final dataset had 416 and 268 records for *B. asper* and *B*. atrox, respectively.

## Environmental layers

We used Bio-climate variables for current conditions from the WorldClim database server (http://www.worldclim.org) [33] at a resolution of approximately 1 km x 1 km. We removed collinear environmental layers with the package *Virtual Species* in R using a threshold of 0.7 over the Pearson's R correlation coefficient [15,34,35]. The selected bioclimatic variables are shown in S1 Table.

## Snake niche modelling score map computation

We predicted species habitat suitability using a maximum entropy algorithm because we used presence-only data [36]. After performing the algorithm described in S1 text, we generated a set of maps representing the 10 best distribution models for each species over each species range to produce habitat suitability maps using a cloglog output format [37]. Next, we removed areas above known altitude thresholds for both species [19,32]. Finally, given that both species are not sympatric [19,32], we combined all the possible permutations of these ten predictions to obtain 100 maps. These score maps are the initial snake niche modeling score ($SNMS^*$).

To select the best $SNMS^*$ map for use in our final snake niche modeling score ($SNMS$), we computed the average value of each $SNMS^*$ map for each department, and we compared it with the logarithm of reported snakebite person-time incidence rate (yearly snakebite

incidence per 100.000 persons, view Eq 2). Reported snakebite risk accounts for snakebite cases in which victims sought medical attention, but it does not differentiate between dry bites and serious cases or among species causing the bite. To compute the reported person-time incidence rate, we averaged SIVIGILA-reported incidence (which corresponds to new yearly cases that received medical attention) between 2010 and 2019, and we normalized it using rural population estimates from the National Administrative Department of Statistics (DANE). Thus, the reported person-time incidence rate corresponds to the averaged snakebite cases per 100.000 population per year collected by SIVIGILA between 2010 and 2019.

To compare each *SNMS** map with reported risk, we computed Pearson's correlation coefficient because our model assumes a linear relationship between both variables (see Eq 2) [38]. Finally, we selected the *SNMS** map with the highest correlation with reported risk as our final envenoming niche modeling score map (*SNMS)*. A standard operating procedure that describes this process is shown in S1 Fig.

## Computation of the accessibility score

We assumed that the distance to the nearest medical center is the only factor affecting antivenom accessibility. Other aspects that can contribute to accessibility are the availability of serum or the presence of trained personnel [6,39,40]. Unfortunately, we did not have access to information on these factors. Hence, we first created a combined map of travel speed using maps of: i) roads and types of roads from the World Food Programme and Open Street maps [41], ii) land coverage from the Food and Agriculture Organization of the United Nations [42], iii) fluvial transport from the Agustin Codazzi institute [43] and *iv)* geographic slope map computed using the *raster* package in R [34,44] and an altitude map from the server WorldClim [33]. The values for travel speeds for each land coverages and type of road is shown in S2 Table. Next, we computed the minimum travel time for each location to the nearest medical center based on the travel speed map obtained by merging input maps. We used a georeferenced dataset of clinics, hospitals, and health services providers from the 2016 Geostatistical National Framework published by the DANE [45]. These three specific health facilities were chosen because they have the capacity to treat snakebite envenoming. This algorithm was performed using the package gDistance in R [46], and the output is a map of the minimum travel time to the nearest medical center.

## Statistical modelling of snakebite underreporting

**Model description.** We based our mathematical model on the hierarchical model proposed by [47]. First, to model the *total* incidence of snakebite, we used the law of mass action because it has the potential to estimate the total incidence of snakebite [17]. Using Eq 2, and writing the model as a generalized lineal model, we can state the following:

$$Ln\left(\frac{\bar{I}_i}{P_i}\right) = \alpha_1 + \alpha_2 SNMS_i + \theta_i \tag{3}$$

Where $\bar{I}$ is the estimation of snakebite *total* incidence, *SNMS* is the averaged snakebite niche modeling score, $\alpha_1$ and $\alpha_2$ are the intersect and the slope of the generalized linear model, respectively; $\theta$ accounts for a normal random noise; and sub-index *i* denotes the geographical scale, in our case municipality. Note that based on data characteristics, our model does not have the capacity to differentiate between an envenoming or a dry bite. Assuming the population as an offset term we can re-write the model as (Eq 4):

$$Ln(\bar{I}_i) = Ln(P_i) + \alpha_1 + \alpha_2 SNMS_i + \theta_i \tag{4}$$

We defined the reporting of snakebite incidence as a counting Poisson process, where the mean is the estimation for *total* incidence (Eq 4) times a reporting fraction $\pi_i$ [47]. Thus, reported snakebite incidence $I^r_i$ can be modelled as:

$$I^r_i = Poisson(\lambda = \pi_i \bar{I}_i) \tag{5}$$

To estimate the reporting fraction, we assumed that it will depend on the proposed accessibility score and a poverty index [5,28]. We used the unsatisfied basic needs index, which is based on access to public services, access to education, household economic dependency, housing conditions, and overcrowding [48,49], as the poverty index. We then defined the accessibility score for each municipality as the geographic-averaged travel time computed previously. By assuming a general linear dependence, the model for the reporting fraction is the following:

$$ln\left(\frac{\pi_i}{1 - \pi_i}\right) = b_1 + b_2 AS_i + b_3 NBI_i \tag{6}$$

We applied the logit function to the reporting fraction $\pi_i$ to guarantee that this fraction will be between 0 and 1. In this part of the model, $b_1$, $b_2$, and $b_3$ are parameters, $AS$ is the accessibility score, and $NBI$ is the poverty index. We restricted $b_1$ and $b_2$ to be negative because the relationship between reporting fraction and accessibility score and poverty index is inverse [5,28]. Thus, the complete model for estimating *total* incidence is as follows (Eq 7):

$$I^r_i = Poisson(\lambda = \pi_i \bar{I}_i)$$
$$Ln(\bar{I}_i) = Ln(P_i) + \alpha_1 + \alpha_2 SNMS_i + \theta_i \tag{7}$$
$$ln\left(\frac{\pi_i}{1 - \pi_i}\right) = b_1 + b_2 AS_i + b_3 NBI_i$$

**Model calibration.**   To estimate the model parameters, we used the MCMC algorithm, which uses a Bayesian approach to fit the model to data [50,51] (prior distributions are shown in S2 Fig). First, the posterior distributions for the parameters were estimated using the NIMBLE package in R [34,50]. Next, we used an automated factor slice sampler (AFSS) to sample the parameters, and then we used four chains from different initial conditions to ensure a global optimal for convergence. Our model converged after 4.7 million iterations; the first 57% were discarded as burn-in, and the following 43% iterations were used to assess convergence by the Gelman-Rubin diagnostic. We used a threshold of non-convergence for this index at 1.1, where values above this threshold indicate non-convergence [52,53].

## Results

### Distribution of venomous snakes

After determining *SNMS*, the selected model for *B. asper* had a train data AUC (Area under ROC curve) of 0.83, a train data CBI (Continuous Boyce Index) of 0.93, and the lowest AICc among all the models. For *B. atrox* the statistics were 0.77 (AUC), 0.97 (CBI), and the difference between its AICc and the model with the minimum AICc (corrected Akaike Information Criterion) was 56. Our model shows that suitability was more spatially heterogeneous for *B. asper* than for *B. atrox*, and the suitability of the latter was greater than 0.55 over its distribution. We also found that the maximum suitability value for *B. asper* was higher than that of *Bothrops atrox*. (Fig 1).

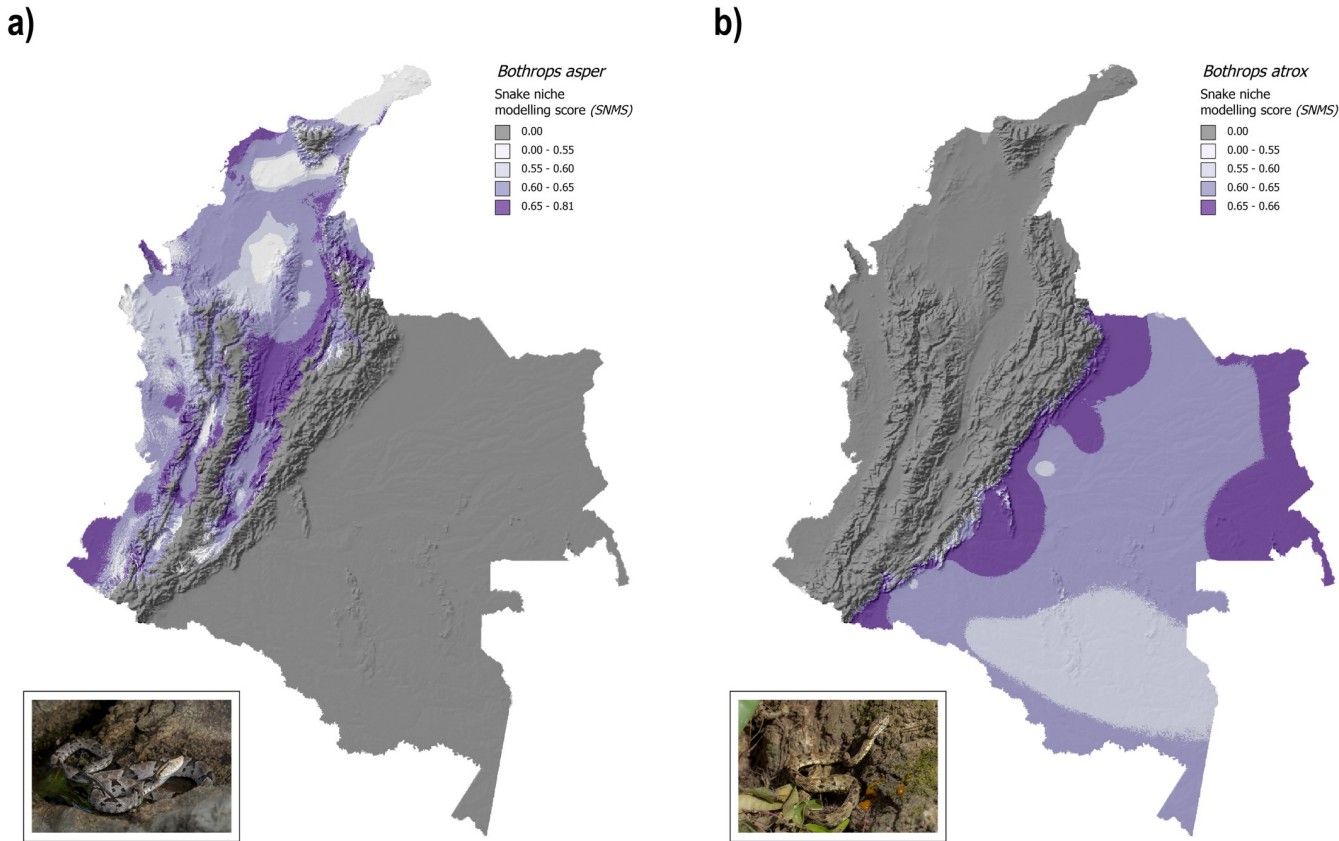

**Fig 1. Snake niche modeling score (SNMS).** Light grey areas are areas outside the range of each species, and dark purple areas are high-suitability areas. Both maps were produced using open-source Geographic Information System Quantum GIS (QGIS). a) Snake niche modeling score for *Bothrops asper*. Note the non-homogeneous distribution of suitability for this species, which is less than 0.55 in light purple areas and the maximum value was 0.81. b) Snake niche modeling score for *B. atrox*. Note that the suitability distribution is more homogeneous for *B. atrox* than for *B. asper*, which was always greater than 0.55 and less than 0.66. Snakes' photographs were taken by author Bravo-Vega. Shaded-relief base map for Colombia was obtained from Natural Earth free vector and raster maps (https://www.naturalearthdata.com/downloads/10m-raster-data/10m-shaded-relief/).

### Envenoming risk score

The logarithm of the SIVIGILA's reported person-time incidence rate was significantly linearly correlated with *SNMS* (Pearson's correlation coefficient: 0.80, p-value < 0.001, Fig 2). This relationship suggests that the main driver of snakebite risk is the abundance or habitat suitability for venomous snakes. In addition, we found that departments where *B. atrox* is distributed form a cluster of high person-time incidence rate values and high values of *SNMS* (View Fig 2), whereas departments where *B. asper* occurs do not form a cluster.

### Underreporting estimation

The maximum index for the Gelman-Rubin diagnostic was 1.01 for $\alpha_1$ and $b_2$, and the multivariate index was 1, indicating convergence [52,53]. Prior and posterior distributions for model parameters are shown in S2 Fig. We estimated an average number of snakebite cases of 5221.56 events per year, whereas SIVIGILA only reported an average of 4689.3 yearly events. As a result, we estimated that 532.26 cases are not reported each year, corresponding to 10.19% of total cases. The geographic distribution of reported cases and our estimation of underreported cases can be seen in Fig 3. Most snakebite cases were reported from Antioquia (average of 683.4 cases annually) (Fig 3A). Thus, the number of cases in which victims do not

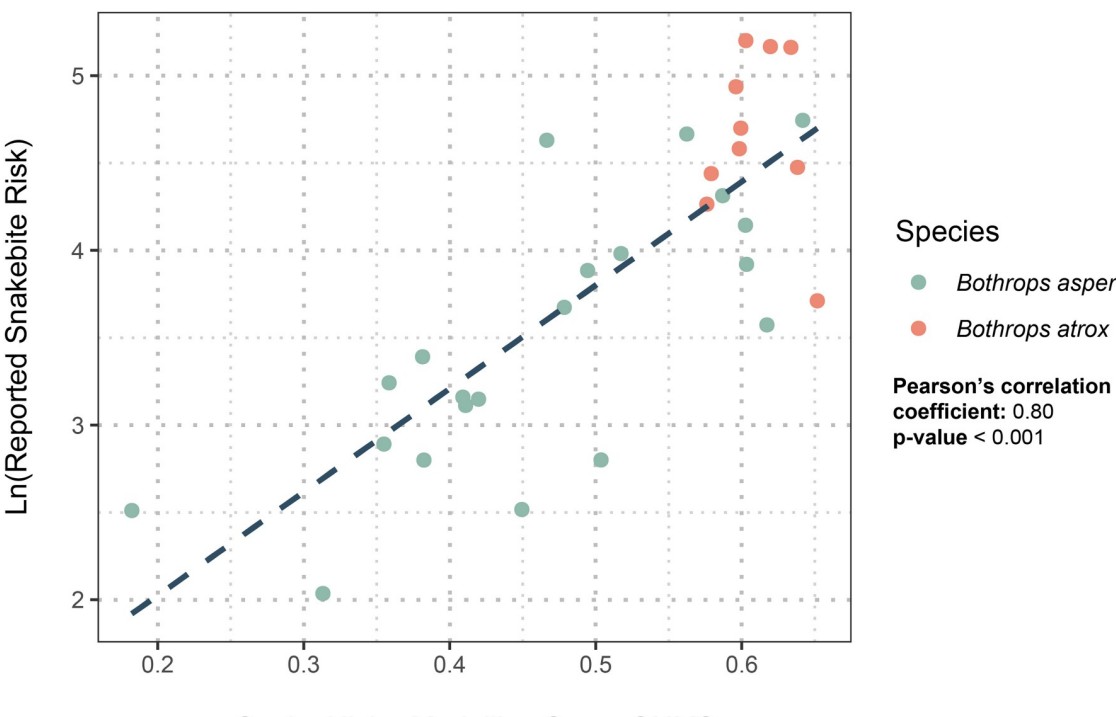

**Fig 2. Linear regression between the logarithm of SIVIGLIA (2010–2019) reported risk and envenoming risk score (SNMS) at the departmental scale.** The Y-axis shows the logarithm of SIVIGILA's reported person-time incidence rate (Envenomings per year per 100.000 persons), and the X-axis shows the average envenoming risk score per department. Each point corresponds to a department. Green circles denote departments where *Bothrops asper* is distributed, and red circles denote departments where *B. atrox* is found.

receive medical attention each year in the country is similar to the number of reported cases in the most affected department. More snakebite cases are reported in departments where *B. asper* is present, but the snakebite risk is higher in departments with *B. atrox* (Figs 2 and 3).

Our underreporting estimates for all municipalities vary between 5.68% and 37.83% of total cases, and this algorithm cannot determine which municipalities have no underreporting (S3 Fig shows the relationship between underreporting, travel time, and unsatisfied basic needs, and the credible intervals for parameters adjustment). Areas located in the eastern and southern parts of the Amazonian and Orinoco regions are the most affected by underreporting, and snakebite risk is highest in these regions (Fig 4). Underreporting percentage is also high along the Pacific and northern Caribbean coast, and these areas are characterized by high poverty rates and poor vial infrastructure (Fig 4) [48,54].

## Discussion

We constructed a mathematical model that estimates snakebite incidence underreporting at a fine political scale. Before exploring the performance and limitations of this model, the inputs required to use this framework include: i) a set of yearly incidence data at the finest administrative level over at least five years (our framework cannot estimate incidence in departments without data, although some modifications can be made to perform a less robust estimation in these areas, as discussed previously in [17]); ii) estimates of the distribution for the most medically significant venomous snake species either through fieldwork [17] or by environmental niche modeling; iii) human population datasets with the same temporal and administrative

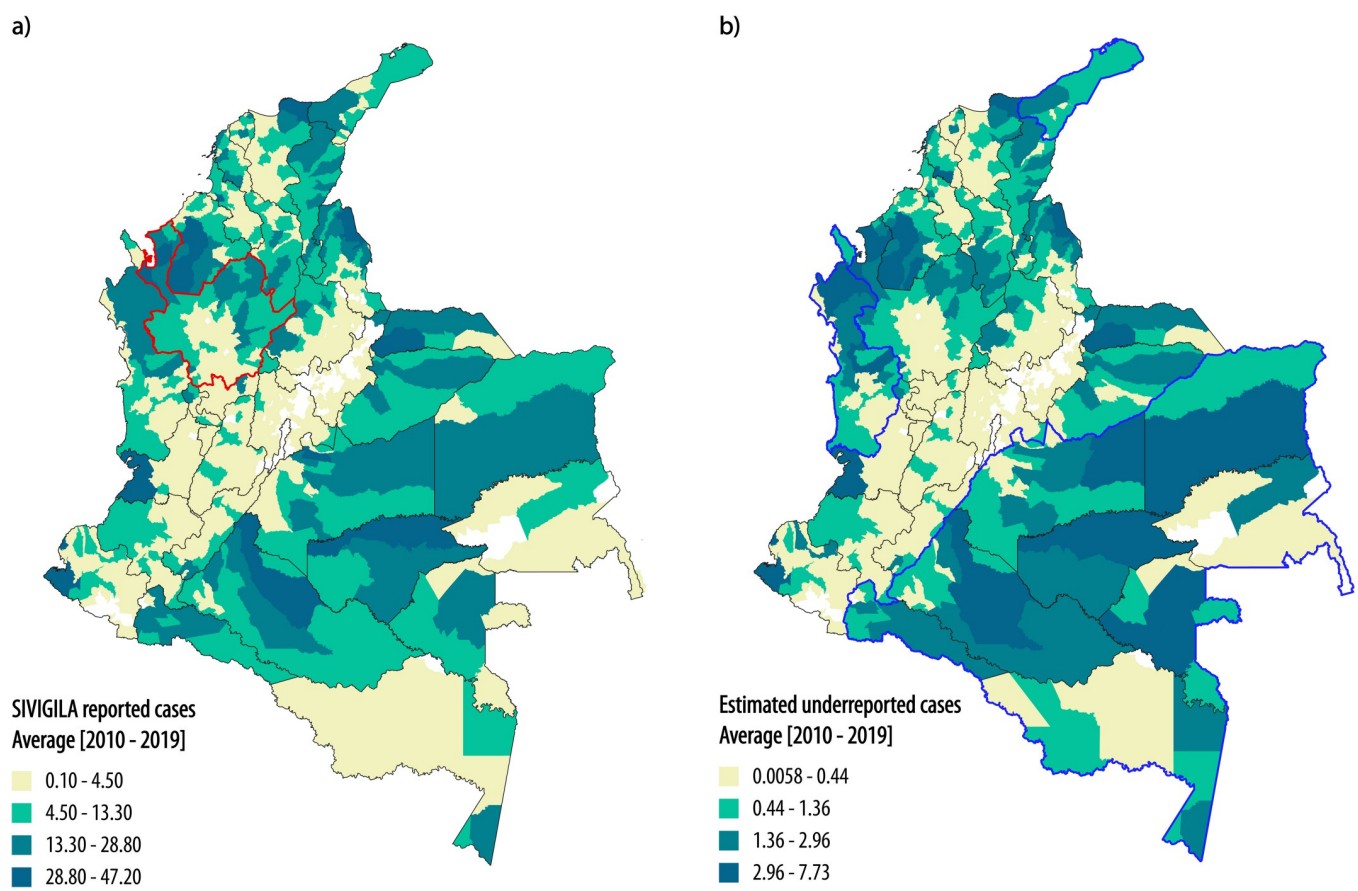

**Fig 3.** a) Yearly snakebite cases reported to SIVIGILA. Note that the spatial distribution of the cases is heterogeneous; most snakebite cases were reported from Antioquia (red border). b) Estimated yearly snakebite cases that are not reported to SIVIGILA. This estimation was done after model parametrization with the MCMC algorithm. Note that the spatial distribution of underreported cases in Antioquia is similar to the distribution of reported cases, but in the south, east, north, and Choco department of the country (blue border), the spatial distribution of cases changes as a consequence of higher percentage of underreporting. Base map of departmental and municipal boundaries of Colombia was obtained from DIVA-GIS free spatial data (https://www.diva-gis.org/datadown).

scale as the incidence data; iv) data on the main roads and fluvial routes, which can be generated by open-source datasets such as Google Maps or from ministries of transport; v) climatic data available in open-source servers such as WordClim or TerraClimate [33,55]; vi) geo-referenced locations of medical centers with the capacity to treat snakebites; and vii) a poverty estimator at the same spatial resolution of the incidence and population data. Since these data are available for several countries, the snakebite burden can be rapidly approximated with our approach.

## Snakebite niche modeling score validity

The estimated distributions for both species (Fig 1) seems adequate because the AUC value is significantly different from 0.5 and close to 1; thus, our model is adequate for differentiating between presence and absence areas, and CBI is positive and close to 1 which means that our predicted distribution is consistent with the presence locations [56,57]. In addition, the values for our models are similar to values reported for other models that have been shown to successfully predict different NTDs distributions [15,16,58,59]. Thus, our distribution maps are reliable because the model statistics indicate that the distribution predictions are robust, and these distributions can map the geographic heterogeneity of the reported snakebite person-time incidence rate (Fig 2).

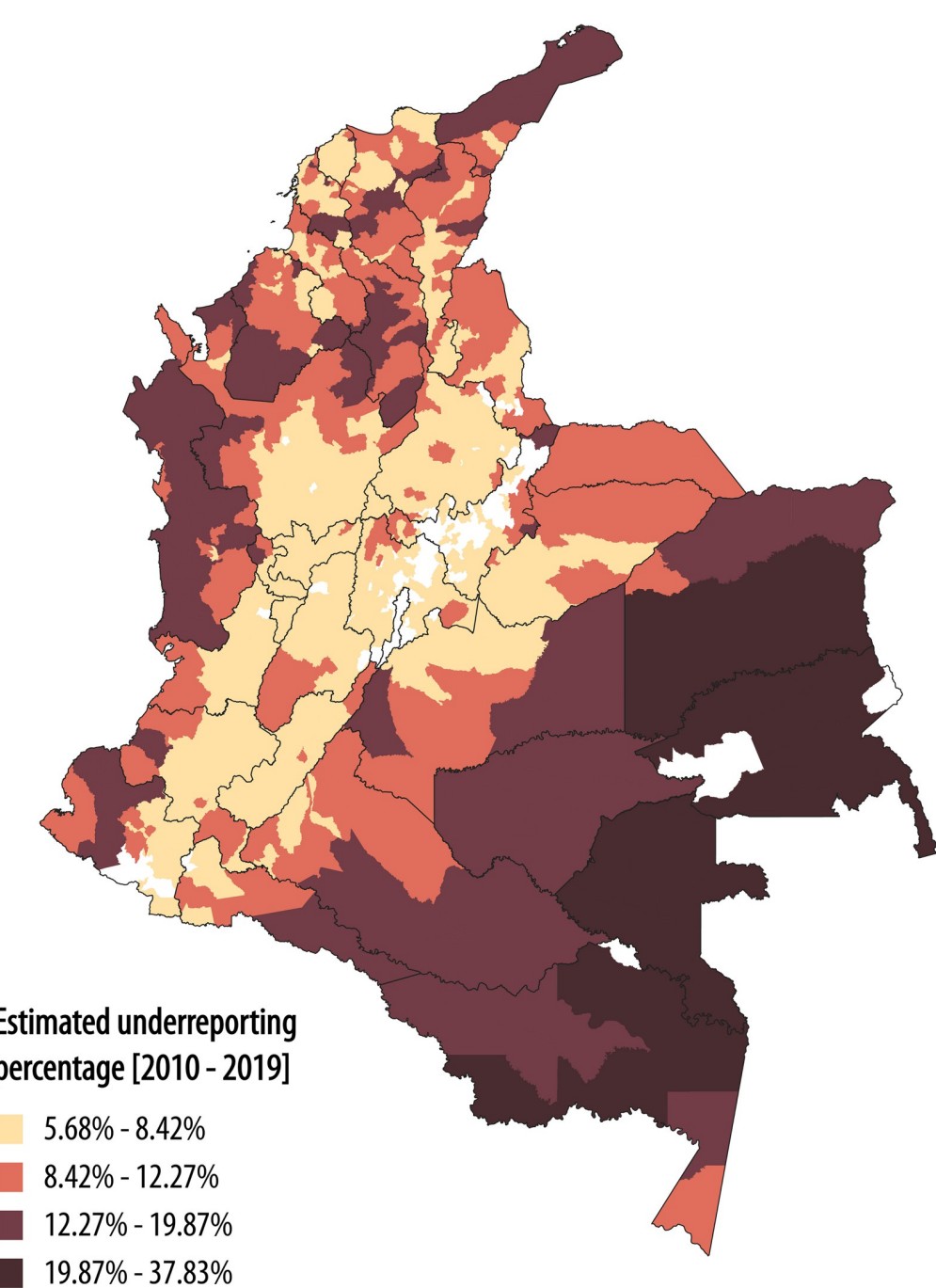

**Estimated underreporting percentage [2010 - 2019]**

- ◻ 5.68% - 8.42%
- ◻ 8.42% - 12.27%
- ◻ 12.27% - 19.87%
- ◻ 19.87% - 37.83%

**Fig 4. Estimated snakebite underreporting.** The areas most affected by underreporting are the southeastern part of Colombia, where *B. atrox* is distributed. Note that the Pacific and northern Caribbean coast region also experiences high underreporting percentage. Base map of departmental and municipal boundaries of Colombia was obtained from DIVA-GIS free spatial data (https://www.diva-gis.org/datadown).

We found that the habitat suitability of both species was higher on humid lowlands, and this is mainly explained by the mean temperature of the warmest quarter for *B. asper* and precipitation of the driest month for *B. atrox* (S3 Table). These environmental variables have not been used to explain snakebite incidence; most studies have used average precipitation and temperature as climatic predictors of snakebite incidence [60–62]. A robust understanding of

the ecology of venomous snakes' is essential for determining the environmental factors associated with snakebite.

## Spatial distribution of snakebite risk in Colombia

The risk of snakebite envenoming is high in Colombia (Fig 1). Snakebite risk is higher in departments with *B. atrox* (Fig 2), where population density, road infrastructure, and urbanization are low, and indigenous populations are high [63–65]. There are two potential hypotheses that can explain this pattern. First, socioeconomic variables can increase snakebite risk: Low urbanization has been demonstrated to be related to snakebite incidence [6,22,66]. In addition, the intensification of anthropogenic activity (e.g., deforestation and expansion of crops lands) can also increase the risk of snakebite [67–69]. In fact, the deforested area is highest in the Amazonian region, and this is mainly caused by fires used to colonize new areas; the third highest deforested area is in the Orinoco region, and this is mainly caused by agriculture [70]. Second, biological factors can increase snakebite risk: *B. atrox* and *B. asper* are different species, thus, differences in their biological attributes might affect the risk that each of them poses to human populations. For example, the abundance of *B. atrox* is higher near rivers in the Brazilian Amazon. Most of the human population in the Colombian Orinoco and Amazon basins is concentrated near streams because these serve as the routes that connect towns [71]. Additionally, the population density of *B. atrox* might be higher than that of *B. asper* because ecological differences between both species, or because *B. asper* is distributed in areas with higher human population density, thus its mortality may be higher and its population density will be lower. Sadly, there are no studies that could enlighten this hypothesis. Finally, feeding ecology studies have shown that *B. atrox* seems to be active during the day more than *B. asper*, thus daily encounters between *B. atrox* and humans may be more frequent [72–75]. These differences might explain the higher snakebite risk in departments with *B. atrox*; however, biological data on these species are absent in Colombia.

We believe that a robust understanding of the natural history of venomous snakes can aid snakebite envenoming risk management [76]. Studies of the trophic ecology and habitat use of these species can provide insights into the micro-habitats that pose the greatest snakebite risks [72,73,77]. In addition, studies of the reproductive activities of these species could clarify temporal variation in snakebite cases [78,79]. Rainfall likely limits snakebite risk in Colombia, and this is consistent with the decreased snakebite incidence in the dry seasons revealed by mathematical models [80]. As such information on venomous snakes is scarce in Colombia, more robust mechanistic hypotheses for the results obtained from epidemiological analyses cannot yet be made [24].

## Underestimation of snakebite threat

The mortality rate of a population due to snakebite can increase substantially without access to antivenom [81,82]. Thus, if cases are underreported, deaths will be critically underestimated. Access to antivenom treatment is not evenly distributed; consequently, regions with low snakebite reporting rates tend to lack access to antivenom [82]. Comparison with field data obtained using surveillance approaches, such as those used in [14,28], is essential for validating our algorithm. The development of novel methodologies is needed to facilitate underreporting estimation, as has been done in [83]. Underreporting field-collected data should be used to finally validate our algorithm, and this can allow its application in several data-deficient areas. Our framework could be used to enhance antivenom availability and decrease mortality.

We estimated that 68% of the country is located more than 2 hours away from the nearest medical center (Rural population in this area = 2'819,166, which corresponds with 24.7% of

total rural population), and 36% of the country is more than 12 hours away from medical centers (Rural population in this area = 422,138, which corresponds with 3.7% of total rural population, View S4 Fig). The long time that it can take snakebite victims to arrive at medical facilities following envenomation increases the snakebite burden and the frequency of underreporting [1,82]. Colombian health authorities have put considerable effort into enhancing the reporting system [22], but there is still a high rate of snakebite underreporting (View Fig 3 and Fig 4): Snakebite is a life-threatening public health problem, and our approach can help quantifying the threat of this NTD.

## Limitations

Although our proposed model can estimate snakebite' incidence using modern mathematical tools, a few limitations of the model require consideration. First, although our risk estimation approach based on snake distribution data makes a significant contribution to achieving public health goals, environmental niche modeling is limited by the quality of the species presence records. Several areas in Colombia cannot be easily accessed, and these areas will invariably have fewer presence records [84]. The maximum entropy algorithm has been shown to be effective for estimating the distributions of species with incomplete presence data [36], but the inclusion of abundance information can improve model reliability.

Second, the assumption that the exposed population comprises the total rural population may not be accurate. Several municipalities have broad elevation gradients, and the exposure risk likely varies throughout municipalities. Additionally, extreme urbanization decreases the abundance of venomous snakes [66,85]. The inclusion of more socio-economic variables could help account for some of this heterogeneity. However, our goal was to keep our model simple by requiring few, relatively accessible sets of input data to estimate underreporting so that it could be easily used in other countries where snakebite is a major public health concern. Our statistical framework utilizes one of the most current tools in model fitting, and it has been used successfully to asses snakebite mortality in India [86]. Nevertheless, our modeling approach has several strengths, it is worth emphasizing that our framework will never substitute a robust surveillance program, which is the responsibility of public health institutions.

## Supporting information

**S1 Fig. Standard operating procedure for selecting the final snake niche modeling score map (SNMS) from the 100 initial snake niche modeling score maps (SNMS\*).**
(DOCX)

**S2 Fig. The prior and posterior distribution for model parameters after fitting convergence.**
(DOCX)

**S3 Fig. Dependence of underreporting on accessibility score and poverty after model parametrization.** As we stated in the model, there is a positive correlation between both variables and underreporting, where low public health coverage and high poverty will increase underreporting fraction. Therefore, our estimation of underreporting varies between 5.68% and 37.83%. Credible intervals were obtained by computing the highest density intervals for posterior distribution.
(DOCX)

**S4 Fig. Travel time to the nearest medical center.** We used georeferenced data of medical centers with the capacity to administrate antivenom, and we determined travel speed with

maps of roads, fluvial routes, slope, and land coverage.
(DOCX)

**S1 Text. Maximum entropy model calibration and selection of the best distribution models.**
(DOCX)

**S1 Table. Environmental layers used for ecological niche modeling.**
(DOCX)

**S2 Table. Travel speeds used for different land coverages.**
(DOCX)

**S3 Table. Variable importance for both species after niche modeling calibration and selection.**
(DOCX)

## Acknowledgments

We thank to the natural history museums of the Universidad Industrial de Santander (UIS), Universidad de los Andes, Institución Universitaria ITM, Universidad del Tolima, Universidad del Valle, Universidad de la Salle (MLS), Instituto Nacional de Salud (INS), Universidad del Magdalena and Universidad Javeriana, as well as all curators of these collections who provided access to specimen records. We also thank Martha Calderón-Espinosa and John D. Lynch (Universidad Nacional de Colombia—ICN), Andres R. Acosta-Galvis (Research institute von Alexander von Humboldt—IAvH), Fernando Sarmiento-Parra and Julieth S. Cardenas-Hincapie (MLS), Martha Patricia Ramírez and Elson Meneses-Pelayo (UIS), Juan Manuel Daza (Universidad de Antioquia—MHUA), and other colleagues who provided us with unpublished specimen records, especially Francisco Javier Ruiz and Juan José Torres (INS) and Sergio Cubides Cubillos (Butantan Institute). We thank Chris Akcali for revising a draft of the manuscript.

## Author Contributions

**Conceptualization:** Carlos Bravo-Vega, Camila Renjifo-Ibañez, Juan Manuel Cordovez.

**Data curation:** Carlos Bravo-Vega, Mauricio Santos-Vega, Leonardo Jose León Nuñez, Teddy Angarita-Sierra.

**Formal analysis:** Carlos Bravo-Vega, Mauricio Santos-Vega.

**Funding acquisition:** Camila Renjifo-Ibañez, Juan Manuel Cordovez.

**Investigation:** Carlos Bravo-Vega, Camila Renjifo-Ibañez, Mauricio Santos-Vega, Juan Manuel Cordovez.

**Methodology:** Carlos Bravo-Vega, Camila Renjifo-Ibañez, Mauricio Santos-Vega, Teddy Angarita-Sierra.

**Project administration:** Camila Renjifo-Ibañez, Juan Manuel Cordovez.

**Resources:** Juan Manuel Cordovez.

**Software:** Mauricio Santos-Vega.

**Supervision:** Camila Renjifo-Ibañez, Juan Manuel Cordovez.

**Validation:** Carlos Bravo-Vega, Mauricio Santos-Vega, Leonardo Jose León Nuñez, Teddy Angarita-Sierra.

**Visualization:** Carlos Bravo-Vega, Leonardo Jose León Nuñez.

**Writing – original draft:** Carlos Bravo-Vega.

**Writing – review & editing:** Carlos Bravo-Vega, Camila Renjifo-Ibañez, Mauricio Santos-Vega, Leonardo Jose León Nuñez, Teddy Angarita-Sierra, Juan Manuel Cordovez.

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
