## [Decision Letter · Decision Letter 0]

12 Oct 2022

Dear Dr Bravo-Vega,

Thank you very much for submitting your manuscript "A generalized framework to estimate snakebite underreporting using statistical models: A study in Colombia" for consideration at PLOS Neglected Tropical Diseases. As with all papers reviewed by the journal, your manuscript was reviewed by members of the editorial board and by several independent reviewers. In light of the reviews (below this email), we would like to invite the resubmission of a significantly-revised version that takes into account the reviewers' comments. 

The topic of this study is relevant in the light of the current global efforts to confront snakebite envenoming. Reviewers appreciated the work while at the same time raised important concerns on many aspects of this manuscript, both in form and content. You should carefully consider all the comments, criticisms and recommendations of the reviewers for preparing a thoroughly revised version.

We cannot make any decision about publication until we have seen the revised manuscript and your response to the reviewers' comments. Your revised manuscript is also likely to be sent to reviewers for further evaluation.

Sincerely,

José María Gutiérrez

Section Editor

Reviewer's Responses to Questions

**Key Review Criteria Required for Acceptance?**

**Methods**

-Are the objectives of the study clearly articulated with a clear testable hypothesis stated?

-Is the study design appropriate to address the stated objectives?

-Is the population clearly described and appropriate for the hypothesis being tested?

-Is the sample size sufficient to ensure adequate power to address the hypothesis being tested?

-Were correct statistical analysis used to support conclusions?

-Are there concerns about ethical or regulatory requirements being met?

Reviewer #1: The manuscript is well written, the theoretical foundation of the model well supported, and I consider that the work deserves to be published. However, a series of observations/suggestions included in the attached pdf file as messages could help to clarify some of the points presented by the authors.

Introduction. This section needs a couple of lines clarifying the factors that can generate underreporting and which of them can be assumed in Colombia. In addition, a generalized mixed effect model is mentioned, with poverty as one of the considered factors. However, there is no previous mention of why this factor should be considered when estimating the incidence of snakebite, nor its relationship with this health problem. I suggest clarifying it in this section.

Reviewer #2: - It would be necessary to ease the understanding explaining what 0.3° or 0.7° mean. Please mention what their equivalents in metric units are. Additionally, it would be necessary to explain how these grid sizes were selected. A reference used for applying the same algorithm (Yañez-Arenas et al., 2014) for a large area from North America to Colombia they used a grid of 0.05°. If this method implies the random removal of all but one observation, why to use such much larger areas to determine SNMS? Why did the authors choose to remove the large majority of their observations? This should be evaluated and improved, or valid limitations clearly explained.

- Relevance of SNMS results for the final under-reporting model:

L255: It is not clear, why if the SNMS is modelling the suitable areas for the two snake species, the “pristine forests” of the amazon region would not be suitable for B. atrox? The goal of Maxent is precisely to predict the expected the distribution of the species based on some observations. Please elaborate on this issue, since that would clearly affect the estimated incidence. In relation to this, the AUC of 0.65 reported for the SNMS of B. atrox shows a very low discrimination power. This value is closer to represent randomness (0.5) than perfect predictability (1.0), thus the validity of SNMS as explanatory variable of the incidence would need to be evaluated, or a better set of variables chosen to produce that factor.

- Validity of the accessibility score:

The authors mention that there was no information on which health facilities were or not able to treat snakebite envenoming. Does it mean that all health facilities of all sizes were considered as able to treat envenoming? This should be clarified as this might result in an over estimation of the treating capacity with additional consequences for under-reporting. Additionally, for the accessibility score, it should be specified what travel speeds were used between different landcover, roads, and rivers, as well as some indication of how those speeds were chosen. All this suggests that the accessibility score might be unrealistic and would lose its representativeness and value.

**Results**

-Does the analysis presented match the analysis plan?

-Are the results clearly and completely presented?

-Are the figures (Tables, Images) of sufficient quality for clarity?

Reviewer #1: The analysis presented match the proposed plan and the results are clear. 

However, implementing the proposed model results in higher estimates of snakebite incidence, 10% of the number recorded in SIVIGILA. However, we are not told which or how many municipalities show this increase. Furthermore, it is unclear if there is a threshold between the model estimates and the recorded incidence data that denotes no underreporting. Or are both estimates compared by some paired test? The authors must explain this point to clarify that the model can effectively identify underreporting cases and discriminate them from cases where there is none.

In Figure 3, the differences between maps 3a and 3b are not notable, possibly due to an effect of the scales of the categories used. This obscures the contrast the authors make in the text about the incidence records and their model estimates, further noted in Figure 4. The authors should consider this point to make their point clearer. In addition, since the Department of Antioquia is mentioned in the text, I suggest they indicate it on the map in Fig 3.

Reviewer #2: (No Response)

**Conclusions**

-Are the conclusions supported by the data presented?

-Are the limitations of analysis clearly described?

-Do the authors discuss how these data can be helpful to advance our understanding of the topic under study?

-Is public health relevance addressed?

Reviewer #1: Conclusions are supported by the data presented. The discussion of the model and its results are pertinent. However, one aspect that is omitted is how to evaluate the results of these estimates empirically. I believe that this section would benefit from advancing how to determine that these underreporting estimates correspond to cases that effectively escape the system. Perhaps the authors have empirical data that allow such an evaluation; failing that, they could leave a possible future study to assess it.

Reviewer #2: (No Response)

**Editorial and Data Presentation Modifications?**

Reviewer #1: As stated above, minor modifications on the manuscript will enhance clarity. Some other comments in attached file. I recommend "Minor revision"

Reviewer #2: Language: 

- Please improve the English throughout the text either by having the manuscript reviewed by a native English speaker or by using an AI editor such as Instatext. 

- Unless you are providing information about which you are completely certain, please avoid using deterministic language that presumes the concepts to be true or definitive.

Referencing: 

- The use of supporting references throughout the paper seems to be unnecessarily bulked-up. There are rather simple ideas with six or more references. This is partially a problem because 50% of the references are 10 or many more years old, and some ideas are supported with what seems counterargument, e.g. L77 where two of the references are actually cross-sectional studies, and the third is a similar study in Costa Rica, where this same idea is mentioned without references.

- L384: Please beware of differences between concepts and opinions/results, and how the former should be referenced, while the latter not.

Basic concepts: 

- In L77-88 is disused the snake diversity leading finally to the selection of two species as main responsible for envenomations. The most recent reference (Nr 37) mentions 64.5% as the proportion of bites by Bothrops. This fundamental concept could be better documented including e.g. a range of percentages, especially considering how old some references are.

- It would also be necessary to clarify whether and how dry bites are considered, since they do not generate envenoming and therefore might influence incidence values if they are recorded or under-reporting if they are not.

- The threshold of 0.9 used to exclude correlated environmental layers is too high. Pearson’s correlations above 0.7 are considered already strong. This should be explained as well as the possible consequences of including highly correlated variables. Following the same example of referenced base articles, Yañez-Arenas et al. 2014, used 0.7 as threshold to remove covariates.

- L170: The description of habitat suitability modelling in text and in ‘S1 text’ is rather vague. The rationale behind the different methodological choices is lacking, especially considering the suggestion of the authors of replication in other countries. Examples of the intermediate results and the basic code to replicate those results could be included in the supplementary material files.

- L180: It is not clear how the reported risk is computed, and as a consequence the difference between reported incidence and reported risk. As presented in the text, reported risk would be an average incidence of several years. If a standardization with the population is used, it should be explained, since incidence is usually already standardized per 100’000 population.

- L183: If the goal is to produce a methodology that can be reproduced, it is necessary to clarify much better the comparison and selection processes done to the SNMS. Please give more detail or ideally create an SOP. Additionally, it is not clear the transit from SNMS to SNMS*.

- L206+: The term ‘real incidence’ is misleading considering that this value is an estimation based on proxy values.

- L219: The poverty index included is clearly an important factor in the computation of incidence, but it is not clear what elements were analysed as part of that index and the references do not explain it, but instead discuss the importance of poverty to snakebite. It would be preferable to keep one or two references indicating the importance of poverty to snakebite, and also a source for the index itself, as well as clarification of the parameters included in it.

- In L258 & 263, is not clear what type of relationship is being suggested, if at all, between areas with low/high number of observations and the resulting SNMS, since both concepts are mentioned simultaneously, and present in high/high, low/high, high/low and low/low combinations in the maps. If the relationship is high/high and low/low, the connection later presented between the SIVIGILA reported risk (L277: SBE/100’000 people/year, which is usually understood as incidence) and the SNMS values would be expected. Overall, these relationships should be explained more clearly.

- L326: Please clarify. In this analysis, was a less robust estimation done in all the places where there was no data? Or, hypothetically no analysis could be done if there were no data at all? The latter would be unnecessary to mention, since no model generates estimates without any data at all.

- L237: For actual replicability, it would be necessary to explain the decisions taken. For instance, why did you choose to use 4.7 million iterations? Is that number justified somehow? 

- L340: The authors address the validity of the risk estimation by mentioning that AUC and BIC values were higher than 0.5. For AUC that statements does not mean much, other than the ‘direction’ of the prediction. For BIC (if it means Bayesian Information Criterion), a value of 0.5 does not represent anything by itself. Please present your results in a more rigorous and precise manner.

- L357-359: the concepts discussed are confusing and imprecise. It has been shown that ‘socio-economic disparities’ e.g. poverty are linked to higher risk of snakebite, and as referenced higher urbanization is linked to lower snakebite incidence and rurality to higher incidence, but low urbanization and intensification of anthropogenic activities are not necessarily linked to socio-economic inequality, as claimed. There are highly urbanized areas where socio-economic inequality is high as well.

Acronyms:

- The acronym SNMS should be better explained. In L135 could be understood that the first S refers to ecological, which of course does not make sense, only in L255 it is clear that it means ‘Snake’.

- In the results, even if they might be well known, please define the acronyms AUC, CBI, AICc, and BIC. Additionally, in L341 it is mentioned once the use of BIC together with AUC. Is CBI or BIC a misspelling of the other?

Equations and formulas:

- The formulas are mislabelled, starting with 3.

- It would be necessary to clarify how � is defined. This is particularly important since the contact rate between different human and snake populations might differ depending on any number of circumstances.

- The paper used to reference the law of mass action, also includes the probability of an encounter resulting in a bite. Why was this omitted in this manuscript?

Plots and figures:

- The 3D plot presented in Fig. S3 shows a very strong influence of accessibility on underreporting, but in comparison, a rather weak effect of poverty. However, that second relationship it is difficult to visualize. It would help to understand the results if numeric results of the models were presented (e.g. parameters and their credible intervals).

- The legend describing the colour code of Fig. 4 talks about the percentage of under reporting, which is mentioned in the text to go from 5.7 to 38.7%, but here indicates 0.057 to 0.366. If this means to show percentages, it should be consistent with the text. Otherwise, it could be misunderstood as 0.057%. Additionally, if as understood, both ranges are the same, the upper limit should be equal in the text and the figure.

- The differences presented in paired maps in figure 3 are minimal and difficult to determine. A complementary approach could be the addition of a map presenting the difference between reported and estimated.

- It is the role of the journal, but I would like to point out how odd is to find results and comments in the legends of figures 1-3, which extend them significantly. I would prefer if the legend were used to describe the figure construction and characteristics.

**Summary and General Comments**

Reviewer #1: (No Response)

Reviewer #2: The article describes an interesting approach to estimate computationally snakebite under-reporting. However, the study present important weaknesses that should be addressed.

PLOS authors have the option to publish the peer review history of their article (what does this mean?). If published, this will include your full peer review and any attached files.

Reviewer #1: Yes: Mahmood Sasa

Reviewer #2: No
---

## [Decision Letter · Decision Letter 1]

16 Jan 2023

Dear Dr Bravo-Vega,

Thank you very much for submitting your manuscript "A generalized framework for estimating snakebite underreporting using statistical models: A study in Colombia" for consideration at PLOS Neglected Tropical Diseases. As with all papers reviewed by the journal, your manuscript was reviewed by members of the editorial board and by several independent reviewers. The reviewers appreciated the attention to an important topic. Based on the reviews, we are likely to accept this manuscript for publication, providing that you modify the manuscript according to the review recommendations. 

The reviewer and the Section Editor are largely satisfied with the revised version of this manuscript. However, there are still some minor points, raised by the reviewer, which need to be taken into consideration for improving the clarity of the manuscript.

Sincerely,

José María Gutiérrez

Section Editor

The reviewer and the Section Editor are largely satisfied with the revised version of this manuscript. However, there are still some minor points, raised by the reviewer, which need to be taken into consideration for improving the clarity of the manuscript.

Reviewer's Responses to Questions

**Key Review Criteria Required for Acceptance?**

**Methods**

-Are the objectives of the study clearly articulated with a clear testable hypothesis stated?

-Is the study design appropriate to address the stated objectives?

-Is the population clearly described and appropriate for the hypothesis being tested?

-Is the sample size sufficient to ensure adequate power to address the hypothesis being tested?

-Were correct statistical analysis used to support conclusions?

-Are there concerns about ethical or regulatory requirements being met?

Reviewer #1: (No Response)

**Results**

-Does the analysis presented match the analysis plan?

-Are the results clearly and completely presented?

-Are the figures (Tables, Images) of sufficient quality for clarity?

Reviewer #1: (No Response)

**Conclusions**

-Are the conclusions supported by the data presented?

-Are the limitations of analysis clearly described?

-Do the authors discuss how these data can be helpful to advance our understanding of the topic under study?

-Is public health relevance addressed?

Reviewer #1: (No Response)

**Editorial and Data Presentation Modifications?**

Reviewer #1: (No Response)

**Summary and General Comments**

Reviewer #1: In this manuscript, the authors present relevant information for understanding snake envenoming, one of the neglected diseases with the most significant impact in the tropics. The information presented is innovative and relevant as a procedure for the statistical analysis of snakebite incidence and estimating underreporting cases is described. The autors apply this approach to Colombia as a case study. This new version has addressed many of this reviewer's initial observations and is more straightforward for PNTD readers. I only have a few minor form suggestions that I outline below. Other than that, I consider the manuscript ready for publication.

Line Reads Suggestion

121 “Our paper describes an algorithm for estimating underreporting using Colombia as a case study” Add a line or paragraph before to introduce the problem of underreporting of the snakebite in Colombia

122-131 These lines are somewhat repetitive of those explained in the methods section. I suggest simply stating that para proposes a statistical framework to estimate snakebite. Consider:

“Our article describes an algorithm to estimate underreporting using Colombia as a case study. For this, we developed a model that considers the risk of snakebites based on the distributions of the most important species from the epidemiological point of view. We tested its performance using the incidence registered in the country's public health system. Using a generalized model incorporating this risk estimator and other factors associated with snakebite incidence, we estimated the reported fraction of total cases and the underreporting of cases on spatial and time scales”.

311 " the multivariate index was 1 (53,54), indicating convergence" . Since the interpretation of the index is described in those references, mark them at the end of the sentence. Ej: the multivariate index was 1, indicating convergence (53,54)

327-330 Consider delineating the regions with the greatest change in Figure 3B 

363-365 " Given that these data are available for several countries, snakebite burden can be approximated quickly with our approach." Consider: Since these data are available for several countries, the snakebite burden can be rapidly approximated with our approach.

396-405 The authors suggest possible differences in density between B. asper and B. atrox, attributing it to differences in their biology. However, differences in density can also result from the relationship with humans: higher mortality rates could be expected in more human-populated environments, that in Colombia roughly coincide with B. asper distribution.

428-430 " We estimated that 68% of the country is located more than two hours away from the nearest medical center, and 36% of the country is more than 12 hours away from medical centers" . But how much does this represent in terms of human population?

PLOS authors have the option to publish the peer review history of their article (what does this mean?). If published, this will include your full peer review and any attached files.

Reviewer #1: Yes: Mahmoud Sasa Marin

Figure Files:

Data Requirements:

Reproducibility:

References

---

## [Editor Report · Decision Letter 2]

20 Jan 2023

Dear Dr Bravo-Vega,

We are pleased to inform you that your manuscript 'A generalized framework for estimating snakebite underreporting using statistical models: A study in Colombia' has been provisionally accepted for publication in PLOS Neglected Tropical Diseases.

Best regards,

José María Gutiérrez

Section Editor

All comments and suggestions raised by the reviewer have been considered for the preparation of the revised version of this manuscript.

---

## [Editor Report · Acceptance letter]

1 Feb 2023

Dear Bravo-Vega,

We are delighted to inform you that your manuscript, "A generalized framework for estimating snakebite underreporting using statistical models: A study in Colombia," has been formally accepted for publication in PLOS Neglected Tropical Diseases.

Best regards,

Shaden Kamhawi

co-Editor-in-Chief

Paul Brindley

co-Editor-in-Chief
